# Neuro-Behavioral Phenotype in 16p11.2 Duplication: A Case Series

**DOI:** 10.3390/children7100190

**Published:** 2020-10-19

**Authors:** Annio Posar, Paola Visconti

**Affiliations:** 1IRCCS Istituto delle Scienze Neurologiche di Bologna, UOSI Disturbi dello Spettro Autistico, 40139 Bologna, Italy; paola.visconti@isnb.it; 2Dipartimento di Scienze Biomediche e Neuromotorie, Università di Bologna, 40126 Bologna, Italy

**Keywords:** autism spectrum disorder, intellectual disability, neuro-behavioral phenotype, genetics, 16p11.2 duplication

## Abstract

Duplications of chromosome 16p11.2, even though rare in the general population, are one of the most frequent known genetic causes of autism spectrum disorder and of other neurodevelopmental disorders. However, data about the neuro-behavioral phenotype of these patients are few. We described a sample of children with duplication of chromosome 16p11.2 focusing on the neuro-behavioral phenotype. The five patients reported presented with very heterogeneous conditions as for characteristics and severity, ranging from a learning disorder in a child with normal intelligence quotient to an autism spectrum disorder associated with an intellectual disability. Our case report underlines the wide heterogeneity of the neuropsychiatric phenotypes associated with a duplication of chromosome 16p11.2. Similarly to other copy number variations that are considered pathogenic, the wide variability of phenotype of chromosome 16p11.2 duplication is probably related to additional risk factors, both genetic and not genetic, often difficult to identify and most likely different from case to case.

## 1. Introduction

Duplications of chromosome 16p11.2 (dup16p11.2), even though rare in the general population (about 0.07% according to Tucker et al., 2013) [1], are one of the most frequent known genetic causes of autism spectrum disorder (about 0.5% according to Levy et al., 2011) [2] and of other neurodevelopmental disorders including developmental delay, intellectual disability, attention deficit hyperactivity disorder (ADHD), developmental coordination disorder, and language disorders [3,4]. Steinman et al. described the neurologic phenotype of dup16p11.2: dysarthria; hypotonia; motor dysrhythmia; microcephaly or macrocephaly; weakness; focal or generalized epilepsy; hyperreflexia or hyporeflexia; eye convergence abnormalities; tremor; tics; electroencephalogram (EEG) focal sharp activity; white matter/corpus callosum abnormalities and ventricular enlargement shown by brain imaging [4]. Regarding psychiatric disorders, in their systematic review, Giaroli et al. found in dup16p11.2 individuals a 14-fold increased risk of psychosis (including schizophrenia, schizoaffective disorder, and bipolar disorder) [5]. Anxiety disorders (including obsessive compulsive disorder) and mood disorders have also been reported [3]. However, data about the neuro-behavioral phenotype of dup16p11.2 carriers are few [6,7]. We dealt with this topic in a sample of pediatric patients.

## 2. Case Presentation

During the period from 2013 to 2020, in the 347 children referred to our Center for a neurodevelopmental disorder (according to the criteria of the Diagnostic and Statistical Manual of Mental Disorders, Fifth Edition: DSM-5) [8] and who have undergone array comparative genomic hybridization (aCGH), we found five (1.4% of the total) dup16p11.2 carriers, all of Caucasian ethnicity. They were four males and one female; two of them were brothers; age at last observation ranged from 7 years 3 months to 15 years 10 months. The size of dup16p11.2 ranged from 59 Kb to 590 Kb. Based on the analysis of segregation in the parents, in four of our patients the duplication was derived from the father, while in one case it was derived from the mother. Only the father of the first two cases had a psychiatric clinical picture attributable to the duplication, while the other parents carrying the duplication were healthy; although intelligence quotient (IQ) assessment was not available, their educational and vocational history were normal.

According to our evaluation, the five patients reported presented with very heterogeneous conditions as for characteristics and severity, from a learning disorder in a child with normal intelligence quotient (IQ) (case 2) to an autism spectrum disorder associated with an intellectual disability (case 1 and 5). Our five patients underwent a neuro-behavioral assessment using standardized tools including Wechsler Intelligence Scale for Children—Fourth Edition (WISC-IV) (cases 2 and 4) or, when language impairment was severe, Leiter International Performance Scale, Revised (Leiter-R) (cases 1, 3, and 5) for evaluating the IQ; “Batteria di Valutazione Neuropsicologica”, an Italian tool used for evaluating cognitive functions such as language, attention, memory, and learning in childhood and adolescence; Child Behavior CheckList (CBCL) (DSM-oriented scales), a questionnaire administered to the parents for evaluating behavior problems; and the Autism Diagnostic Observation Schedule—Second Edition (ADOS-2) [9], the gold standard tool for the evaluation of social-communicative competences and range of interests and behaviors when an autism spectrum disorder is suspected. Module 1 of ADOS-2 was administered in cases 1, 3, and 5; module 2 in case 4; and module 3 in case 2. We considered also the calibrated severity score of ADOS-2, a measure of autism symptom severity ranging from 1 (corresponding to a typical development) to 10 (maximum severity of autism). Informed consent, covering also the anonymous publication of case reports, was obtained from the patients’ parents.

Table 1 and Table 2 describe the main genetic, clinical, instrumental, and in particular the neuro-behavioral features of each of the five individuals with dup16p11.2 we reported. Figure 1 represents the pedigree of the family of cases 1 and 2.

Some comments on the description of the five cases follow. For cases 1, 3, and 4, the learning difficulties we reported reflect lower overall cognitive functioning and are not to be considered specific. Concerning the epileptic seizures of cases 1 and 3, unfortunately, based on the available clinical and electrophysiological data of our patients, a syndromic classification is not possible, also because the semiology of the seizures is apparently not in line with the features of the EEG. The paroxysmal events reported for case 3 after the age of 7 years 9 months are probably not epileptic; a possible alternative diagnostic hypothesis is that of a periodic paralysis.

## 3. Discussion

Our case report confirms the wide heterogeneity of the clinical neuropsychiatric phenotypes associated with dup16p11.2, ranging from an at least apparently normal condition (see in particular the three healthy parents who were carriers of the same duplication of the affected sons) to a severe impairment of neurodevelopment such as an autism spectrum disorder associated with an intellectual disability, with a series of intermediate conditions including an intellectual disability combined with developmental disorders outside the autism spectrum. These findings led us to formulate some hypotheses in order to explain the clinical heterogeneity of dup16p11.2 carriers. First of all, one involved variable could be the size of the duplication. However, based on the analysis of our sample, this variable does not seem to be determining since case 5, carrier of the smallest duplication (only 59 Kb, partially overlapping with the duplications found in case 1, 2, and 4), is one of the two individuals with the greatest impairment, being affected by autism spectrum disorder plus intellectual disability. Another variable underlying this clinical heterogeneity could be the concomitant presence of another copy number variation (CNV) detected by the aCGH, according to the so-called two-hit hypothesis. The only associated CNV involving genes that we found is a deletion of 7q31.1 in case 5, which includes intron 5 of inner mitochondrial membrane peptidase 2-like (*IMMP2L)* gene, whose mutations/deletions have been associated with several neuropsychiatric disorders including Gilles de la Tourette syndrome, ADHD, developmental delay, and also autism spectrum disorder [10]. It cannot be excluded that this CNV might have contributed to the clinical picture of case 5, also considering the motor tics he presented. Moreover, it should be stressed that additional genetic factors modifying the dup16p11.2 clinical picture could be represented by mutations of single genes that can be highlighted by the whole-exome sequencing, not by the aCGH, and that have not been considered in this paper. It should also be emphasized that even the deletions that do not involve genes (see case 4 and case 5) could play a role in the expression of the phenotype, due to their possible function of regulating genes that are located elsewhere [11]. Finally, a further variable underlying the clinical heterogeneity of dup16p11.2, as for the CNVs in general, could be represented by the concomitant presence of environmental factors, which of course are somewhat difficult to study due to the very high number of factors that could theoretically be involved: pre- and perinatal hypoxic-ischemic suffering, early exposure to environmental pollutants and to endocrine-disrupting chemicals, just to give some possible examples [12].

As a final consideration, we underline the presence in all our patients of heterogeneous sleep disorders and the lack in our cases of specific facial and body dysmorphisms. The first finding could further suggest a link between neurodevelopmental disorders and 16p11.2 duplication, since sleep disorders are an early clinical feature frequently reported in children with neurodevelopmental disorders. The lack of specific dysmorphisms, which is a feature common to many CNVs, once again suggests the importance of carrying out genetic tests such as the array-CGH even in cases in which the phenotype is characterized solely by a neurodevelopmental disorder, in the absence of somatic peculiarities.

We are well aware of the limitations of this study due to the low number of our case series, however we believe that the considerations originating from the analysis of our patients could stimulate the scientific debate about the neuro-behavioral phenotype in 16p11.2 duplication, also because in literature data specifically about this topic are few.

In conclusion, similarly to other CNVs that are considered pathogenic, the wide variability of dup16p11.2 phenotype, ranging from a normal condition to severe impairment of the neurodevelopment, according to Green Snyder et al. [3], is probably related to additional risk factors, both genetic (including associated CNVs or mutations of single genes) and not genetic (i.e., environmental: see above), often difficult to identify and most likely different from case to case.

## Figures and Tables

**Figure 1 children-07-00190-f001:**
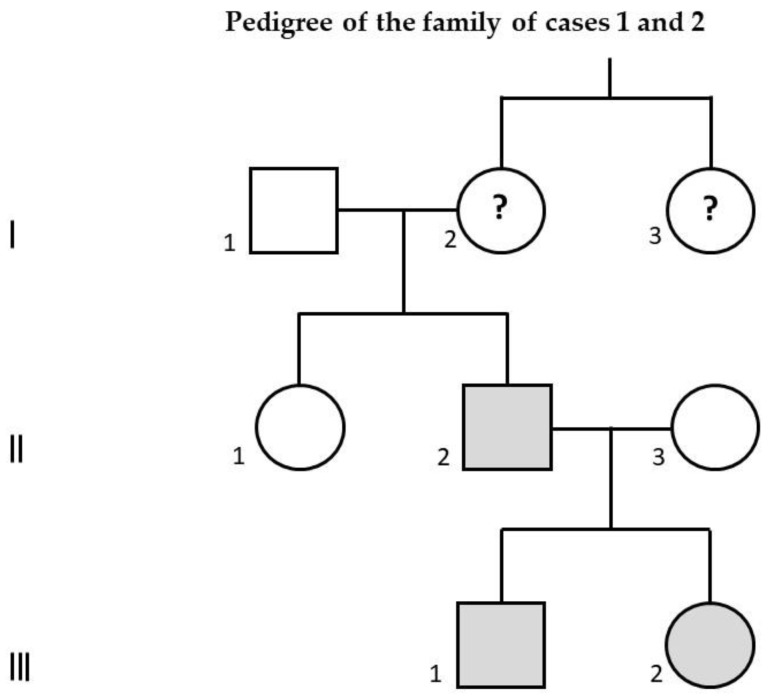
Family pedigree of case 1 (III-1), affected by autism spectrum disorder associated with intellectual disability, and of case 2 (III-2), affected by learning disability. Their father (II-2), presenting with the same 16p11.2 duplication, is affected by schizophrenia. His aunt (I-3) is also affected by schizophrenia: in her, the 16p11.2 duplication has not been documented and therefore it is only a hypothesis; if she were really a carrier of this duplication, evidently the paternal grandmother of cases 1 and 2 (I-2) would be a healthy carrier of 16p11.2 duplication. Gray color: confirmed carriers of 16p11.2 duplication; question mark: hypothetical carriers of 16p11.2 duplication.

**Table 1 children-07-00190-t001:** Genetic and clinical features of five cases with dup16p11.2.

Cases: Sex and Age at last Observation	aCGH Features	Family Antecedents	Pre-, Peri-Neonatal Period	Psychomotor Development	Sleep	Neurological Examination	Movement Disorders	Dysmorphisms
1. Male, 12 years 10 months	(average spatial resolution ~130 Kb): dup16p11.2 (size: 457 Kb; nucleotides involved: 29,673,954–30,131,105), inherited from the father.	Brother of case 2. Father, carrier of the same 16p11.2 duplication: schizophrenia. Father’s aunt: schizophrenia.	Cesarean section for breech presentation.	Language and motor delay. Age of walking: 17 months. First words: 12 months; first sentences: ~36 months.	Difficulty falling asleep, awakenings, nightmares, pavor nocturnus.	Clumsiness, toe-walking, severe language impairment, echolalia. HC: ~25th percentile.	Stereotypies with upper limbs.	Testicle retained. Weight: ~25th percentile.
2. Female, 11 years	As case 1 (see cell above).	Sister of case 1 (see above).	Normal.	Normal. Age of walking: 12 months. First words: 13 months.	Restless sleep.	Normal. HC: ~50th percentile.	Absent.	Very protruding ears. Weight: ~50th percentile.
3. Male, 12 years 10 months	(average spatial resolution ~100 Kb): dup16p11.2 (size: 590 Kb; nucleotides involved: 28,543–29,133), inherited from the healthy father.	Twin sister: intellectual disability. Mother: sleepwalking.	Twin pregnancy with threatened abortions; at 35–36 weeks of gestation cesarean section for breech presentation.	Language delay (prevailing in the production). Age of walking: 15 months. First words: 36 months.	At first reduced sleep times; later restless sleep and perhaps sleepwalking.	Very poor speech. HC: ~25th percentile.	Absent.	Absent. Weight: ~25th percentile.
4. Male, 15 years 10 months	(average spatial resolution ~100 Kb): (1) dup16p11.2 (size: 525 Kb; nucleotides involved: 29,673,754–30,198,753), including *KIF22*, *PRRT2*, and *ALDOA* genes, inherited from the healthy mother. (2) del7q31.1 (size: 60 Kb), not involving genes, inherited from the healthy mother.	Neoplasms in paternal line. Intellectual disability and psychiatric disorders in two cousins of the mother.	Normal.	Language delay. Age of walking: 13–14 months. First words: 12 months; first sentences: after 36 months.	Difficulty sleeping. Nocturnal awakenings.	Right eye exophoria. HC: ~25th percentile.	Absent.	Retrognathia, thin nose, pectus excavatum, thin fingers, flat pronated feet, thin skin. Weight deficit: <3rd percentile.
5. Male, 7 years 3 months	(average spatial resolution ~25 Kb): (1) dup16p11.2 (size: 59 Kb; nucleotides involved: 29,652,999–29,712,097), including *SPN* and *QPRT* genes, inherited from the healthy father. (2) del7q31.1 (size: 29 Kb) including intron 5 of *IMMP2L* gene, inherited from the healthy father. (3) del9p24.3 (size: 66 Kb), not involving genes, inherited from the healthy mother.	Personality disorder: maternal uncle. Alzheimer’s disease: paternal grandmother.	FIVET. Initial biovular twinning with spontaneous interruption of biovularity at the 3rd month of gestation. Hypovalid sucking.	Language delay. Age of walking: 15 months. First words: 12 months; first sentences: ~36 months.	Nocturnal awakenings in the first 2 years of life.	Echolalia. Unintelligible speech. HC: ~75th percentile.	Motor tics. Bruxism. Stereotypies (iterative hops).	Absent. Weight: ~25th percentile.

aCGH: array comparative genomic hybridization; *ALDOA*: aldolase, fructose-bisphosphate A; dup16p11.2: duplication of chromosome 16p11.2; FIVET: fertilization in vitro and embryo transfer; HC: head circumference; *IMMP2L*: inner mitochondrial membrane peptidase 2-like; Kb: kilobases; *KIF22*: kinesin family member 22; *PRRT2*: proline-rich transmembrane protein 2; *QPRT*: quinolinate phosphoribosyltransferase; *SPN*: sialophorin.

**Table 2 children-07-00190-t002:** Neuro-behavioral, clinical, and instrumental features of five cases with dup16p11.2.

Cases: Sex and Age at last Observation	Intellectual Functioning	Neuropsychological Assessment	Psychiatric and Behavioral Assessment	Epilepsy and Other Paroxysmal Events	EEG Findings	Brain MRI Findings	Medical Comorbidity
1. Male, 12 years 10 months	Moderate intellectual disability. Nonverbal IQ = 40.	Language: phonological alterations. Severe learning disorder in reading, writing, and mathematics. Good visual memory.	Deficits of social communication and social interaction; restricted and repetitive behaviors, interests or activities; sensory abnormalities: autism spectrum disorder (severity level 3: requiring very substantial support, according to DSM-5). Irritability (auto-aggressiveness). Hyperactivity. Attention deficit. ADOS-2: autism; CSS: 8. CBCL: above the clinical threshold for Attention Deficit/Hyperactivity Problems.	Since the age of 7 years, absence seizures; cluster of myoclonic seizures at 9 years and 8 months. No further seizures thereafter. Therapy: lamotrigine.	Multifocal (left parieto-temporal and right fronto-temporal) and diffuse paroxysmal abnormalities at 7 years 9 months. Afterwards, only slow activities prevailing on the right and after on the left hemisphere.	Dilated perivascular spaces in the left retrotrigonal white matter and at the junction between the anterior third and the middle third of the corpus callosum.	None.
2. Female, 11 years	Normal. FSIQ = 99, ICV = 98, IRP = 106, IML = 97, IVE = 94.	Learning disorder in reading, writing, and mathematics.	Hyperactivity. Poor tolerance of frustration. ADOS-2: out of autism spectrum; CSS: 1. CBCL: below the clinical threshold in all areas.	None.	Not performed.	Not performed.	None.
3. Male, 12 years 10 months	Moderate intellectual disability. Nonverbal IQ = 44.	Learning: severe impairment of reading, writing and, above all, mathematics.	Oppositional defiant disorder. Hyperactivity. Attention deficit. Disinhibition. Poor frustration tolerance. Hetero-aggressiveness. Depressive and anxious symptoms. Difficulty relating to peers. Therapy: risperidone. ADOS-2: out of autism spectrum; CSS: 2. CBCL: above the clinical threshold for Affective Problems, Anxiety Problems, Attention Deficit/Hyperactivity Problems, and Oppositional Defiant Problems.	At 7 years 9 months, one generalized tonic vibratory seizure. After, some prolonged paroxysmal events with falls, pain at lower limbs, inability to walk, and preserved consciousness (ictal EEG: normal): periodic paralysis?	At 5 years 8 months, right temporal paroxysmal abnormalities during wake and sleep. Afterwards, normal.	Mild dilatation of lateral ventricles.	Laryngospasm. Polyallergic individual.
4. Male, 15 years 10 months	Mild intellectual disability. FSIQ = 68, ICV = 70, IRP = 69, IML = 79, IVE = 91.	Learning problems (reading, writing, and mathematics). Impairment of verbal memory.	Oppositional defiant behavior. Depression. Anxiety. Phobia for insects. Attention deficit. Problematic relationships with peers. Introversion. Impulsiveness. Hetero-aggressiveness. Poor tolerance of frustration. Persecutory ideas. Important dependence on adult reference figures. ADOS-2: out of autism spectrum; CSS: 2. CBCL: above the clinical threshold for Affective Problems, Anxiety Problems, and Oppositional Defiant Problems.	None.	Normal in wake and sleep.	Reduced thickness of the pituitary gland and dilatation of some perivascular spaces in the retrotrigonal area.	None.
5. Male, 7 years 3 months	Mild intellectual disability. Nonverbal IQ = 70.	Language: phonological alterations. Unable to draw. Recognizes letters and numbers. Temporo-spatial disorientation. Deficit of fine motor skills. Visual memory above normal.	Deficits of social communication and social interaction; restricted and repetitive behaviors, interests or activities; sensory abnormalities: autism spectrum disorder (severity level 2: requiring substantial support, according to DSM-5). ADOS-2: autism; CSS: 6. CBCL: above the clinical threshold for Attention Deficit/Hyperactivity Problems.	None.	During wakefulness, slow spike-waves in the left posterior regions at 4 years 5 months; normal at 7 years 3 months.	Normal.	Frequent upper respiratory tract infections in the first 2 years of life.

ADOS-2: Autism Diagnostic Observation Schedule—Second Edition; CBCL: Child Behavior CheckList; CSS: calibrated severity score; DSM-5: Diagnostic and Statistical Manual of Mental Disorders, 5th ed.; EEG: electroencephalogram; FSIQ: full-scale intelligence quotient; IQ: intelligence quotient; MRI: magnetic resonance imaging; PRI: perceptual reasoning index; PSI: processing speed index; VCI: verbal comprehension index; WMI: working memory index.

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
