# Peer review of "Neuro-Behavioral Phenotype in 16p11.2 Duplication: A Case Series"

_children, 2020, doi:10.3390/children7100190_

Round 1
Reviewer 1 Report
This case study of the clinical presentation of 16p11.2 duplication carriers is of great clinical importance and the case study series is clearly laid out. Advances in genetic testing mean that children are increasingly receiving diagnoses such as 16p11.2 duplication, and therefore case studies such as this manuscript are vital for expanding the knowledge base of genetic conditions such as 16p11.2 duplication. The authors should be thanked for collecting detailed phenotype information on this relatively rare condition, this area of research is often complex and time-consuming.
Introduction
- “Duplications of chromosome 16p11.2 (dup16p11.2), even though rare in the general population (about 0.07% according to Tucker et al., 2013) [1], are one of the most frequent known genetic causes of autism spectrum disorder and of other neurodevelopmental disorders including developmental delay, intellectual disability, attention deficit hyperactivity disorder (ADHD), developmental coordination disorder, and language disorders [2,3]”
It would be great if the authors could add in the frequency of 16p11.2 duplication in autism spectrum disorder to help the reader.
- “However, detailed reports about the neuro-behavioral phenotype of dup16p11.2 carriers are lacking.”
As much as I agree that more detailed reports are needed, I think the authors should make this sentence less bold as there have been large international consortia studies that have examined the neurobehavioural phenotype of 16p11.2 duplication (PMID: 26629640, PMID: 30664628). The authors should consider adding in these references to the manuscript, though I should note a conflict of interest in that I was involved in one of these consortia, however I do believe these are currently the largest published studies.
Case presentation
- The authors mention that consent was gained, though could they clarify that consent specifically covered the publication of case reports.
- For the parent carriers whose clinical presentation appeared healthy, it would be interesting to know if IQ or educational & vocational history is available, as this information would provide insight into the parents cognitive functioning.
Discussion
- “In conclusion, similarly to other CNVs that are considered pathogenic, the wide variability of dup16p11.2 phenotype, according to Green Snyder et al. [2], is probably related to additional risk factors, both genetic and not genetic, often difficult to identify and in all probability different from case to case.”
There are a lot of important concepts in this last sentence, and I feel the discussion would benefit from these concepts being expanded on further. Could the authors expand on what “not genetic” factors could be, and why these factors are often difficult to identify. Also, personally I would rephrase “in all probability different from case to case” to something less colloquial.
Author Response
Responses to Reviewer 1

Reviewer 2 Report
The authors report 5 cases of 16p11.2 duplication with various breakpoints.
An extensive literature exists about 16p11.2 copy number variants and their broad phenotype spectrum. These articles describe large cohorts of carriers clinically ascertained, which gives an overall picture of the main phenotypic features (Cf D'Angelo et al., JAMA 2016) but, on the other hand, does not bring out the finer points or comprehensive medical history. The analysis of small cohorts therefore has the advantage of a finer description closer to clinical routine practice which can be useful to clinicians. Therefore, globally a more detailed picture of each case would be of great value to illustrate the numerous developmental trajectories related to 16p11.2 dup.
- Introduction: There is a long list of signs here that could be summarised to make reading easier (line 34-39).
- Results:
Methods: What kind of Weschler scale was used? Was it always the same for each case? Please detail as well as the IQ score(s).
Was it a full scale IQ? Subtest for verbal and non verbal IQ would be interesting, since severe speech and language disorders have been reported associated to 16p11.2 dup/del.
- The tables are not easy to read.
- Table 1: Specify, if possible, age of walking and of first words.
16p11.2 dup were associated to lower head circumference (HC) and low weight, it would be interesting to have HC and weight for each case, in Z-score (standard deviation).
Case 5: Rather than "incomprehensible speech", the term "unintelligible speech" seems more appropriate if we are talking about a phonological problem.
For "family history" column, pedigrees would be easier to read. In the "neurological examination" column there are mainly developmental findings, clinical examination being almost normal in every cases.
a column dedicated to speech and language development might be more relevant, and the "pure" neurological findings summarized in the text.
Brain MRI findings are also not very relevant, since we have non specific and common findings with "Virchow Robin" spaces dilation.
- Table 2: Please indicate IQ score in "intellectual functioning".
For cases 1, 3 and 4, learning difficulties are noted but all theses cases displayed intellectual disability: therefore it raises the question whether these problems are really specific disorders or they reflect a lower overall cognitive functioning? Please justify.
Epilepsy: is there a possible syndromic classification? Apart "benign epilepsy with centrotemporal spikes (so called rolandic epilepsy)", 16p11.2 dup/del have not been associated to specific epilepsy syndromes; please comment on this. For cases 1 and 3 the semiology of the seizures does not seem to be in line with the EEG features (e.g. absence seizures and multifocal paroxysmal anomalies ), again a a comment would be useful. In case 3, the description of paroxysmal episodes unclear? It seems to non epileptic but what was the differential diagnosis?
A this step, authors should draw a more complete/detailed picture of each patients, this would add an important value to their report.
Reviewer 3 Report
I have reviewed this article from my expertise as a non clinician- biomedical and molecular scientist with a background in neurodevelopmental disorders.
The authors present a report of the broad clinical presentation of 5 Dup16p11.2 cases diagnosed by aCGH.
They present a good characterization of genomic findings and regions involved, with a focus on the neuro-behavioral presentation. I consider the report is a good resource for clinicians since includes a good representation of a range of cases: long and small duplication, inheritance from parents, siblings, and cases with additional genomic rearrangements.
Here I mention some minor concerns to address:
- Cite appropiate reference for ADOS-2 and could additionally specify modules performed.
- Specify cases for which ADOS-2 was not performed.
- Gene names should be italicized as per nomenclature guidelines for human genes.
- Change word introne for intron.
- Adding references for environmental factors will be more informative in the discussion.
Suggestions for improving understanding of the cases and discussion:
A) Perhaps a schematic representation of the location of the duplications can help to understand whether the duplicated regions in the cases presented here are contained within another one, overlapped, distal, etc.
B) Line 89, perhaps clarify if the region in case 5 is contained within the duplications of the other cases, since, this could just be the critical region for the phenotype.
C) Discussion: Two cases have additional genomic rearrangements. The discussion about this characteristics focuses on the IMMP2L intron, but it should also suggest the presence of regulatory regions within the deletions. Even though these deletions do not involve genes, they may represent regulatory regions and also the advances in the field of gene regulation show that noncoding regions may impact gene expression elsewhere. Therefore there is still a possibility that some of the phenotypes seen in such cases may be influenced by lacking a putative complex trans regulation from the deleted regions. Since this article may be a resource for clinicians it is important to not clinically discard a priori a rearranged region because it does not contain genes.
Round 2
Reviewer 2 Report
Further details have been added to give a better picture of 16p11.2 carriers.
Author Response
All recently modified parts (Round 2) of the paper have been highlighted in blue. All previously modified parts (Round 1) have been highlighted in green.
Reviewer 2 Comments:
Figure 1. The means of reviewer 2 is to ask for a better picture of figure 1 which can explain 16p11.2 carriers more clear.
Response: We have clarified in Figure 1 and in its caption (line 95-96) who the confirmed carriers of the 16p11.2 duplication and the hypothetical ones are.